# LEARNING TO REPRESENT AND PREDICT IMAGE SEQUENCES VIA POLAR STRAIGHTENING

## ABSTRACT

Observer motion and continuous deformations of objects and textures imbue natural videos with distinct temporal structures, enabling partial prediction of future frames from past ones. Conventional methods first estimate local motion, or optic flow, and then use it to predict future frames by warping and copying content. Here, we explore a more direct methodology, in which frames are mapped into a learned representation space where the structure of temporal evolution is more readily accessible. Motivated by the geometry of the Fourier shift theorem and its group-theoretic generalization, we formulate a simple architecture that represents video frames in learned polar coordinates to facilitate prediction. Specifically, we construct networks in which pairs of convolutional channel coefficients are interpreted as complex-valued, and are expected to evolve with slowly varying amplitudes and linearly advancing phases. We train these models on next-frame prediction, and compare their performance with that of conventional methods using optic flow, and other learned predictive networks, evaluated on natural videos from two datasets. We find that the polar predictor achieves high prediction performance while remaining interpretable and fast, thereby demonstrating the potential of a flow-free video processing methodology that is trained end-to-end to predict natural video content.

## 1 INTRODUCTION

One way to frame the fundamental problem of vision is that of representing the signal in a form that is more useful for performing visual tasks, be they estimation, recognition, or guiding motor actions. Perhaps the most general "task" is that of temporal prediction, which has been proposed as a fundamental goal for unsupervised learning of visual representations (Földiák, 1991). But previous research along these lines has generally focused on estimating these transformations rather than using them to predict: extracting slow features (Wiskott & Sejnowski, 2002), and finding dictionaries and sparse codes that have slow amplitudes and phases (Cadieu & Olshausen, 2012).

In video processing and computer vision, a common strategy for temporal prediction is to first estimate local translational motion, and to then copy and paste content to predict the next frame. Such motion compensation is an important component in making compression schemes like MPEG successful (Wiegand et al., 2003). These video coding standards are the fruit of major engineering efforts, they make digital video communication feasible and are widely used. But motion estimation is a difficult nonlinear problem, and existing methods fail in regions where temporal evolution is not translational. For example, in cases of expanding or rotating motion, discontinuous motion at occlusion boundaries, or mixtures of motion arising from semi-transparent surfaces (e.g., viewing the world through a dirty pane of glass). In compression schemes, these failures of motion estimation lead to prediction errors, which are then fixed by sending additional corrective bits.

Human perception does not seem to suffer from such failures - at least, our subjective sense is that we can anticipate the time-evolution of visual input even in the vicinity of these commonly occurring non-translational changes. In fact, those changes are often the most informative ones as they reveal object boundaries, provide ordinal depth and other information about the visual scene. This may imply that humans make use of a different strategy, perhaps bypassing altogether the estimation of motion, to represent and predict evolving visual input. Toward this end, and inspired by recent hypotheses that primate visual representations support prediction by "straightening" the temporal

trajectories of naturally-occurring input (Hénaff et al., 2019), we formulate an objective for learning an image representation that facilitates prediction by linearizing the temporal trajectories of frames of natural video.

This separation of the instantaneous representation and the temporal prediction is best motivated by considering the behavior of rigidly translating video content when viewed in the frequency domain. First in section 1.1, we review how translation corresponds to steady phase advancement in the frequency domain, and then in section 1.2, we explain how this relationship reduces prediction of rigidly translating content to angular extrapolation. We place this observation in the general context of group representation theory in section 1.3. Next in section 2, we describe how to fit parameterized mappings of individual video frames into complex coefficients which can be temporally predicted by phase advancement. These predicted representations are then used to synthesize an estimated frame, and the entire systems are trained end-to-end to minimize next frame prediction errors. In section 3, we report training results of several such systems, and show that they produce systematic improvements in predictive performance over conventional motion compensation methods, or direct predictive neural networks. Finally, in section 4, we relate our approach to existing work and then in section 5 discuss its significance and implications.

## 1.1 BASE CASE: THE FOURIER SHIFT THEOREM

Our approach is motivated by the well-known behavior of Fourier representations with respect to signal translation. Specifically, the complex exponentials that make up the Fourier basis are the eigenfunctions of the translation operator, and translation of inputs produces systematic phase advances of frequency coefficients. Let $x \in \mathbb{R}^N$ be a discrete signal indexed by spatial location $n \in [0, N-1]$, and let $\widetilde{x} \in \mathbb{C}^N$ be its Fourier transform indexed by $k \in [0, N-1]$. We write $x^v(n) = x(n-v)$, the translation of $x$ by $v$ modulo $N$ (ie. circular shift with period N). Defining $\phi = e^{i2\pi/N}$, the primitive N-th root of unity, we can express the Fourier shift theorem as:

$$\widetilde{x^v}(k) = \sum_{n=0}^{N-1} x(n-v)\phi^{-kn} = \sum_{m=-v}^{N-1-v} x(m)\phi^{-km}\phi^{-kv} = \phi^{-kv}\sum_{n=0}^{N-1} x(n)\phi^{-kn} = \phi^{-kv}\widetilde{x}(k).$$

This relationship may be depicted in a compact diagram:

$$
\begin{array}{ccc}
\widetilde{x}(k) & \xrightarrow{\text{advance phase}} & \phi^{-kv}\widetilde{x}(k) \\
\Big\uparrow{\scriptstyle\mathcal{F}} & & \Big\downarrow{\scriptstyle\mathcal{F}^{-1}} \\
x(n) & \xrightarrow{\text{shift}} & x(n-v)
\end{array}
\tag{1}
$$

where $\mathcal{F}$ indicates the Fourier transform. In the context of our goals, the diagram illustrates the point that transforming to the Fourier domain renders translation a "simpler" operation: a phase advance is a rotation in the two dimensional (complex) plane.

## 1.2 PREDICTION VIA ANGULAR EXTRAPOLATION

Now consider observations of a signal that translates at a constant velocity over time, $x(n,t) = y(n - vt)$. Although the temporal evolution is easy to describe, the trajectory of the signal is quite complicated, rendering prediction difficult. As an example, Figure 1 shows a signal consisting of a sum of two sinusoidal components. Transforming the signal to the Fourier domain simplifies the description. In particular, the translational motion now corresponds to circular motion of the two (complex-valued) Fourier coefficients associated with the constituent sinusoids.

The motion is further simplified by a polar coordinate transform to extract phase and amplitude of each Fourier coefficient. Specifically, the motion is now along a straight trajectory, with both phases advancing linearly (but at different rates), and both amplitudes constant. Note that this is a geometric property that holds for any rigidly translating signal, and offers a simple means of predicting content over time. Indeed, we can use the shift property on $x(n, t+1) = x^v(n, t)$ and observe that prediction

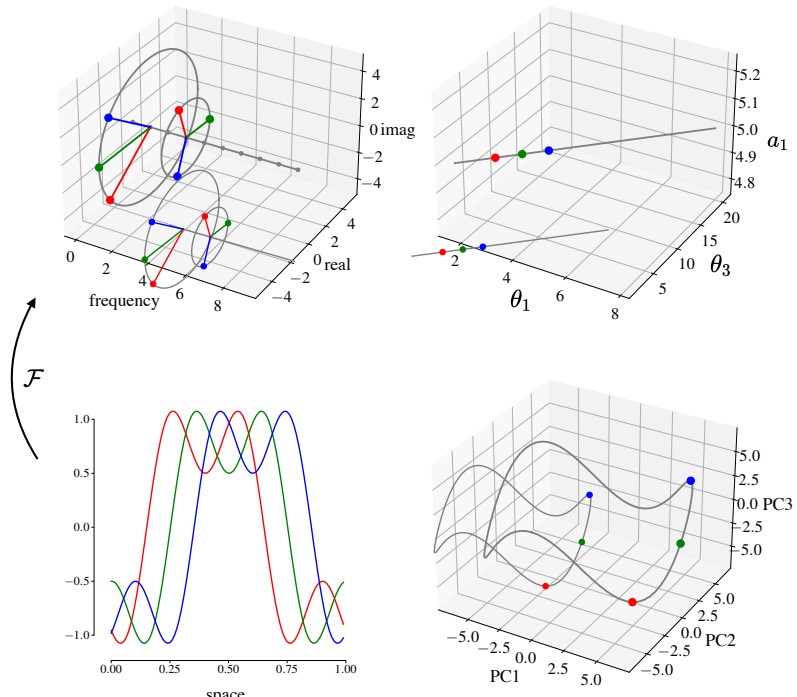

Figure 1: Translation of a 1D signal consisting of a sum of two sinusoidal components: $x(n, t) = \sin(2\pi(n-t)) + \sin(2\pi 3(n-t))/2$. Lower left: three snapshots of the signal as it translates. Lower right: In the high-dimensional space representing the signal (each axis corresponding to the signal value at one location), the temporal trajectory is highly curved. Shown is the projection of the signal vector into the 3D space of the top three principal components. Three colored points indicate the tree snapshots in lower left panel. Upper left: Fourier transform of the signal, showing complex-valued coefficients as a function of frequency. In this representation the temporal trajectory corresponds to linearly increasing phase of the two sinusoidal components, each at a rate proportional to its frequency. Upper right: a polar coordinate transform to amplitude and phase of each frequency component leads to a representation that evolves along a straight line, and is thus readily predictable (phases are unwrapped for display purposes).

is now reduced to linear extrapolation of each coefficient's phase. We have the three step process:

$$\widetilde{x}(k, t) = \sum_{n=0}^{N-1} \phi^{-kn} x(n, t), \qquad \text{(analyze)}$$

$$\widetilde{x}(k, t+1) = \phi^{-kv} \widetilde{x}(k, t), \qquad \text{(advance phase)}$$

$$x(n, t+1) = \frac{1}{N} \sum_{k=0}^{N-1} \phi^{kn} \widetilde{x}(k, t+1). \qquad \text{(synthesize)}$$

Since we assume that the motion from time $t$ to $t+1$ is identical to that from time $t-1$ to $t$, the phase advance $kv$ can be computed from the past two representations as $kv = \angle \widetilde{x}(k, t) - \angle \widetilde{x}(k, t-1)$, where $\angle z$ is the phase of the complex number $z$. Thus, a polar coordinate transformation in the Fourier domain converts translational motion into straight trajectories which are predictable via linear phase extrapolation.

## 1.3 GENERALIZATION: REPRESENTING COMMUTATIVE CONTINUOUS GROUPS

Fourier analysis can be seen as a special case of the representation theory of compact commutative Lie groups (Mackey, 1980). In harmonic analysis, the celebrated Peter-Weyl Theorem (1927) guarantees the existence of harmonic basis functions that generalize the Fourier shift property for

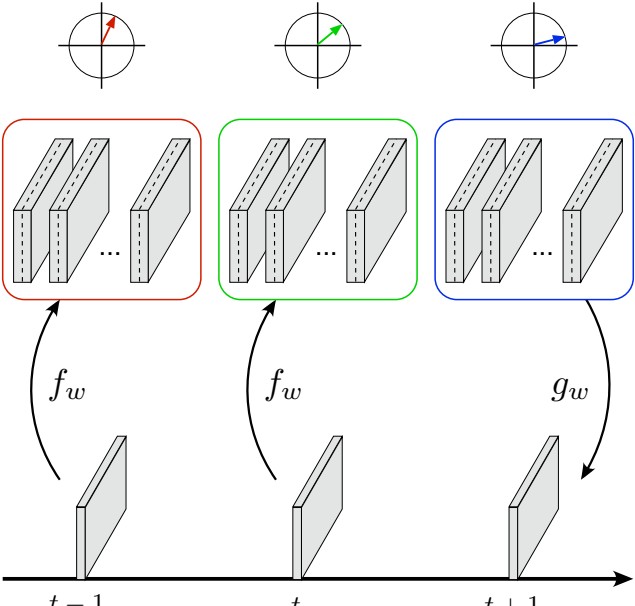

Figure 2: Unsupervised predictive representation learning framework. Each frame is transformed using a parametric mapping $f_w$, to an internal space consisting of pairs of spatial channels. Predictions of individual complex coefficients at time $t + 1$ are computed by advancing their phase by an amount equal to the phase advance over the interval from $t - 1$ to $t$ (at each time step, one such coefficient is depicted as a vector in two dimensions). Predicted frames are then constructed by applying the parameterized inverse mapping $g_w$ on the advanced coefficients. Both forward and inverse mappings are trained to minimize the prediction error (typically, the mean squared error between the predicted and actual frame at $t + 1$).

arbitrary compact Lie groups. Specifically, the linear unitary representation of G is given by an explicit complete orthogonal basis, constructed from the finite-dimensional irreducible representations of G (Hall, 2013). The defining property of an irreducible representation is that its span is invariant under group action, so that arbitrary shifts can be expressed as linear combination of basis functions (an example is the construction of steerable filters (Freeman et al., 1991) in the computational vision literature). In the case of compact commutative Lie groups, complex irreducible representations are one-dimensional, meaning that harmonic basis functions come in pairs. Therefore, the angular extrapolation mechanism of the previous section can be employed for prediction in a much wider setting than that of translational motion. Since the groups at play in natural videos (and their corresponding harmonic basis functions) are unknown, we aim to learn them from data.

## 2 LEARNING TO PREDICT WITH ANGULAR EXTRAPOLATION

To generalize beyond translation and the Fourier transform, we aim to *learn* a representation of video frames that enables prediction via angular extrapolation. Specifically, we focus on next frame prediction, and optimize two parameterized mappings: one for the analysis and one for the synthesis transform. This framework is illustrated in Figure 2, notice its similarity with diagram 1.

### 2.1 ARCHITECTURE AND OBJECTIVE FUNCTION

When focusing on a small region in an image sequence, the transformation observed as time passes can be approximated as a *local* translation. That is to say, in a spatial neighborhood around position $n$, we have: $x(n, t + 1) \approx x(n - v, t)$. We can reuse the setup described for global rigid translation, replacing the Fourier transform with a local convolutional operator (Fleet & Jepson, 1990), processing each spatial neighborhood of the image independently and in parallel, and applying angular extrapolation to the coefficients computed at each position.

Analogous to the Fourier transform, we use the same weights for the encoding and decoding stages, that is to say the analysis operator is the transpose of the synthesis operator. Sharing these weights reduces the number of parameters and simplifies interpretation of the learned solution. This "polar predictor" (hereafter **PP**) is consistent with the general scheme described in figure 2 where $f_w$ is taken to be linear and convolutional, and $g_w$ is its transpose. In practice, we assumed 64 convolutional channels with filters of size $17 \times 17$ pixels, with no additive constants.

At every position in the image (spatial indices are omitted for clarity of notation), each coefficient $y_j(t)$ is computed as an inner product between the input $x(t)$ and the filter weights $w_j$ of each channel $j \in [0, 63]$ : $y_j(t) = w_j^T x(t)$. In order to obtain phases, we combine coefficients in pairs, indexed by $k \in [0, 31]$, which can be written as single complex coefficient as: $z_k(t) = y_{2k}(t) + iy_{2k+1}(t) \in \mathbb{C}$, and expressed in polar coordinates: $z_k(t) = a_k(t)e^{i\theta_k(t)}$. This polar coordinate transformation is the only non-linear step used in the polar predictor architecture, it pairs of coefficients and differs from more typical activation functions which operate on each coefficient independently.

With this notation, linear phase extrapolation reduces to $\hat{z}_k(t+1) = a_k(t)e^{i(\theta_k(t)+\Delta\theta_k(t))}$, where the phase advance $\Delta\theta_k(t)$ is equal to the phase difference over the interval from $t-1$ to $t$: $\Delta\theta_k(t) = \theta_k(t) - \theta_k(t-1)$. This can be written in a more compact form, using complex arithmetic, as:

$$\hat{z}_k(t+1) = \frac{z_k(t)^2 \overline{z_k(t-1)}}{|z_k(t)||z_k(t-1)|}, \tag{2}$$

where $\overline{z}$ denotes complex conjugation and $|z|$ the complex modulus. Note that the intermediate phase variable drops-out when the whole computation is expressed using complex coefficients. This formulation bundles together phase and amplitude computations and has the benefit of handling phases implicitly, bypassing the phase unwrapping difficulty and avoiding the notorious instability of angular variables (indeed phase is unstable when amplitude is low), while still computing the desired angular extrapolation. In practice, this phase advance can be implemented using real valued elements only, as:

$$\begin{bmatrix} \hat{y}_{2k}(t+1) \\ \hat{y}_{2k+1}(t+1) \end{bmatrix} = \begin{bmatrix} \cos\Delta\theta_k(t) & -\sin\Delta\theta_k(t) \\ \sin\Delta\theta_k(t) & \cos\Delta\theta_k(t) \end{bmatrix} \begin{bmatrix} y_{2k}(t) \\ y_{2k+1}(t) \end{bmatrix}, \tag{3}$$

where trigonometric identities can be used to calculate $\cos\Delta\theta_k(t)$ and $\sin\Delta\theta_k(t)$ without explicitly evaluating phases. We find that such an indirect formulation of phase processing is necessary for the stability of training, as previously noted in the texture modeling literature (Portilla & Simoncelli, 2000).

As a second generalization we use a non-linear representation implemented with deep convolutional neural networks for both for the encoder $f_w$ and the decoder $g_w$. The goal is to learn a transformation in which the coefficients can be predicted using the polar extrapolation mechanism. This "deep polar predictor" (**deepPP**) model can be written as $\hat{x}(t+1) = g_w(A(t)f_w(x(t)))$, where $A(t)$ depends on both $x(t)$ and $x(t-1)$ and implements angular extrapolation in the code space as described in equation 3. This model amounts to a signal factorization into local amplitude and phase in a non-linear representation. We aim to study the effect of non-linearity separately from that of spatial scale, therefore we chose the number of layers and the kernel sizes to match the effective receptive field size of the shallow polar predictor, that is to say units in both models operate on the same size input region in the image. In practice, both the encoder and the decoder are composed of 4 convolutional layers, each with 64 channels, and using filter kernels of size $5\times5$ followed by half-wave rectification (ReLU). In this architecture, the encoder and the decoder are not tied together.

The convolutional kernels $w$ are learned by minimizing the average prediction error, measured as squared error:

$$\min_w \mathbb{E}_t \|x(t+1) - \hat{x}(t+1)\|_2^2.$$

We restrict the computation of this prediction error to the center of the image because moving content that enters from outside the video frame is inherently unpredictable. In practice, we trim a 17-pixel strip from each side. Note that we only perform valid convolutions to avoid artificial interference with prediction, indeed zero-padding creates undesirable boundary artifacts.

## 2.2 COMPARISON MODELS

We compare our method to the traditional motion compensated coding approach that forms the core of inter-picture coding in well established compression standards such as MPEG. Block matching is an essential component of these standards, allowing the compression of video content by three orders of magnitude with moderate loss of information. For each block in a frame, typical coders search for the most similar spatially displaced block in the previous frame (typically measured with MSE), and communicate the displacement coordinates to allow prediction of frame content by translating blocks of the (already transmitted) previous frame. Specifically, we implemented a "diamond search" algorithm (Zhu & Ma, 2000) operating on blocks of $8 \times 8$ pixels, with a maximal search distance of 8 pixels which balances accuracy of motion estimates and speed of estimation (the search step is computationally intensive). Here, we use the estimated displacements to perform causal motion compensation (**cMC**), using displacement vectors estimated from the last two observed frames ($x_{t-1}$ and $x_t$) to predict the *next* frame ($x_{t+1}$) rather than the current one (as in MPEG).

To isolate the effects of the prediction in polar coordinates, we also implemented a predictor that uses linear extrapolation on the responses of a deep neural network (**deepL**), with architecture identical to that of the deep polar predictor. That is to say, we replace equation 3 by: $\hat{y}_j(t+1) = 2y_j(t) - y_j(t-1)$, which amounts to enforcing linear dynamics in the latent space of a non-linear representation.

We also implemented a more direct convolutional neural network predictor (**CNN**), that maps pairs of observed frames to an estimate of the next frame (Mathieu et al., 2016). Such a mapping jointly transforms and predicts visual signals without explicitly partitioning spatial content representation and temporal feature extrapolation. Specifically we used a CNN composed of 20 stages, each consisting of 64 channels, and computed with $3 \times 3$ filters without additive constants, followed by half-wave rectification. Unlike the previous architectures, this model jointly processes pairs of frames.

## 2.3 DATASETS AND TRAINING

To train, test and compare these models, we use the DAVIS dataset (Pont-Tuset et al., 2017), which was originally designed as a benchmark for video object segmentation. Image sequences in this dataset contain diverse motion of scenes and objects (at different magnitudes and speeds, with fixed and moving camera, etc.), which make next frame prediction challenging. Each clip is sampled at 25 frames per second, and is approximately 3 seconds long. The set is subdivided into 60 training videos (4741 frames) and 30 test videos (2591 frames). We pre-processed the data, converting all frames to monochrome luminance values, and scaling their range to the interval $[-1, 1]$. Frames are reduced by cropping a $256 \times 256$ central region, where most of the motion tends to occur, and by down-sampling by a factor of two down to size $128 \times 128$ pixels. We hypothesize the temporal evolution present in natural signals to be sufficiently diverse, therefore we do not apply any data augmentation procedures. We crop videos temporally into brief snippets of 11 frames, which allows for prediction of 9 frames, and process them in batches of size 4. We train each model for one hundred epochs using the Adam optimizer with default parameters and a learning rate of $3 \cdot 10^{-4}$ (Kingma & Ba, 2015). The learning rate is halved at epochs 50, 60, 70, 80, 90, 100. We use batch normalization with no additive terms before every half-wave rectification, rescaling by the standard deviation of channel coefficients. Similarly, we also trained on the larger UCF-101 dataset (Soomro et al., 2012). This dataset, initially designed for action recognition, contains about 2.5 million frames, which amounts to over 27 hours of video data. Note that the clips are only available in compressed formats and may contain motion artifacts (due to inter-frame coding), unlike the DAVIS dataset that is available as individual frames. We used the same pre-processing procedure, except that we only reduced by cropping to a $128 \times 128$ central region, without any down-sampling. We employ the same training procedure, except that we only run training for 25 epochs.

## 3 UNSUPERVISED REPRESENTATION LEARNING

### 3.1 PERFORMANCE

We summarize the prediction results in Table 1.First, observe that the predictive algorithms considered in this study perform significantly better than the baseline obtained by simply copying the last frame. Second, the polar predictor (PP) performs about as well as the vanilla convolutional neural

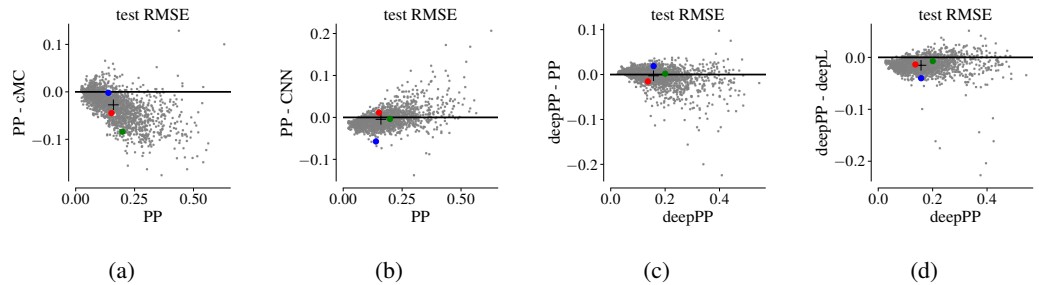

Figure 3: Detailed performance comparison (in Root Mean Squared Error) of predictive algorithms. Each point corresponds to a frame in the test set. Vertical axes represent difference in performance: For points lying below the horizontal axis, the method whose performance is plotted on the horizontal axis achieves a lower RMSE than the comparison method. Black crosses indicate average RMSE. Red point corresponds to example in Figure 4, green to Figure 5 and blue to Figure 7.

network (CNN) in terms of test mean squared error on DAVIS. This is remarkable because the PP model has roughly 30 times fewer parameters and uses a single non-linearity, while the CNN is composed of 20 non-linear layers. This shows that using a polar non-linearity contributes to performance. Finally, observe that deepPP achieves the lowest mean squared error, notably it outperforms the deepL model which is identical in architecture but employs a linear prediction mechanism. It also outperforms the CNN on the UCF-101 test dataset while remaining significantly simpler. This is further evidence that polar coordinates provide an inductive bias beneficial to the prediction task.

Table 1: Prediction error computed on the DAVIS and UCF-101 datasets. Values indicate average Mean Squared Error.

| dataset | split | Algorithm | | | | | |
|---------|-------|------|------|-------|-------|-------|--------|
|         |       | Copy | cMC  | deepL | CNN   | PP    | deepPP |
| DAVIS   | train | 0.064 | 0.048 | 0.034 | 0.031 | 0.036 | **0.028** |
|         | test  | 0.065 | 0.049 | 0.037 | 0.035 | 0.035 | **0.032** |
| UCF-101 | train | 0.0302 | –      | 0.0220 | **0.0210** | 0.0245 | 0.0216 |
|         | test  | 0.0286 | 0.0299 | 0.0217 | 0.0215 | 0.0229 | **0.0210** |
| # param |       | 0 | 2 | 665,856 | 666,496 | 18,496 | 665,856 |

While average performance values provide a compact summary, it is also informative to examine the distribution of prediction errors on individual frames from the test set. Figure 3 shows pairwise comparison of the predictive algorithms for each frame in the DAVIS dataset. To make the contrast more apparent, we display performance difference on the vertical axis. Note that while the models have been optimized to reduce mean squared error, we show root mean squared error (RMSE) in order to facilitate visual inspection of the results (the concavity of the square root spreads out small differences). We see that (a) The polar predictor representation systematically outperforms causal motion compensation, especially on difficult examples. (b) The polar predictor outperforms the CNN on the bulk of easy to medium cases but this advantage is reversed for harder examples. (c) The deep polar predictor outperforms the single layer polar predictor overall, indicating that the non-linearity in representation can help, but there are many exceptions. (d) The deep polar predictor clearly outperforms the deep linear predictor, revealing the strong benefit of using a polar extrapolation mechanism over a linear one.

## 3.2 LEARNED FILTERS

In order to better understand these results, we visualized the learned filters of the polar predictor trained on the DAVIS dataset and observed that the learned filter are Gabor-like (selective for orientation and spatial frequency) and that that they tile the frequency domain. Interestingly, filters in a

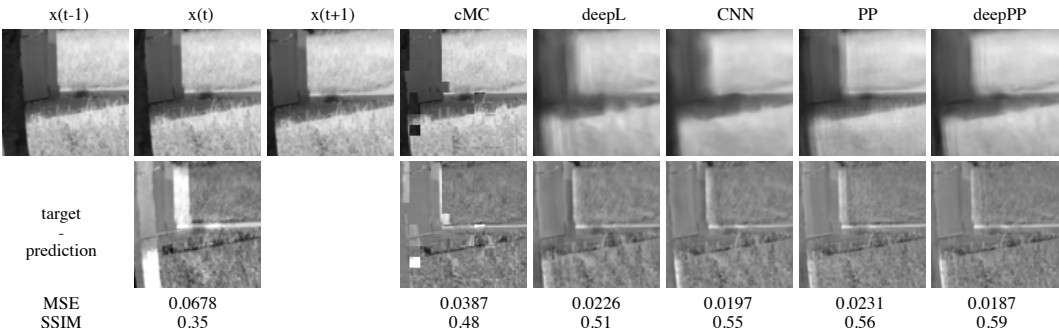

Figure 4: A typical example image sequence from the DAVIS test set. The first three frames on the top row display the unprocessed images, and last five frames show the respective prediction for each method (with their shorthand above). The bottom row displays error maps computed as the difference between the target image $x(t + 1)$ and each predicted next frame on the corresponding position in the first row. Images, predictions and error maps are all shown on the same scale.

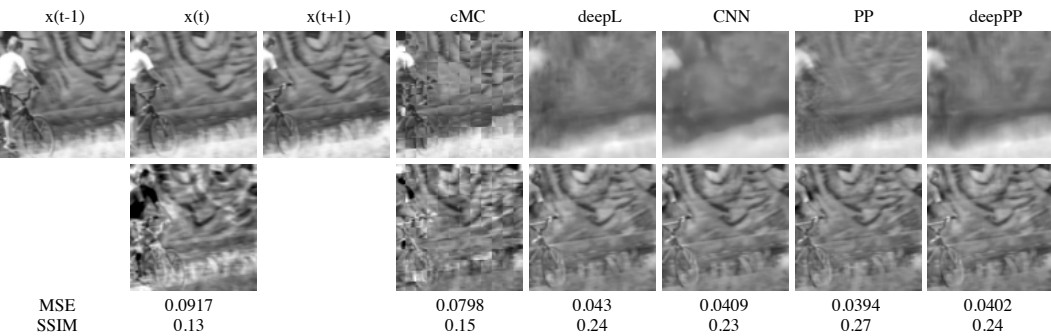

Figure 5: Another example image sequence - see caption of Fig. 4

pair have a similar frequency amplitude and, in the spatial domain, they are related by a 90 degrees phase shift. In other words, optimizing for angular extrapolation produces pairs filters that are in quadrature, see in Figure 9a and 9b in the Appendix. This relationship is analogous to that of sines and cosines and is consistent with the structure of the angular extrapolation described in equation 3.

### 3.3 EXAMPLES

Consider a set of example videos, chosen to illustrate behaviors of the methods being compared. In Figure 4, we see a wall, its shadow and their sharp boundaries on a grass background as the camera moves. Both PP and deepPP generate good results, cMC produces a sharp prediction, but with significant blocking artifacts, and both the CNN and deepL tend toward excessive blurring. Figure 5 shows an example with nonrigid motions/deformations. As the biker advances to the right, the camera tracks and leads its displacement. Here, the cMC is is significantly sharper than the others, but introduces substantial artifacts. Again, the PP methods produce sharper results than either the CNN or deepL methods. A couple additional informative examples are displayed in the appendix.

## 4 RELATED WORK

Our method is conceptually related to sparse coding with complex-valued representations (Cadieu & Olshausen, 2012) in that it factorizes natural videos into form and motion. But it differs in a number of important ways: (1) sparse coding focuses on representing, not predicting, the signal; (2) we do not promote sparsity of either amplitude or phase components; (3) finally, the discontinuity arising from selection of sparse subsets of coefficients seems at odds with the representation of continuous group actions, while our explicit mapping into polar coefficients aims for a smooth and continuous parameterization of the transformations that occur in natural videos. Several other studies have

aimed to learn representations that decompose signal identity and attribute (ie. a *what-where*, or *invariance-equivariance* factorization). In particular learning linearized features from video was explored using a heuristic extrapolation mechanism on a toy dataset (Goroshin et al., 2015). The authors developed specialized "soft max-pooling" and "soft argmax-pooling" modules while we rely on principles of signal processing and representation theory to chose the polar non-linearity and describe an effective and stable implementation thereof.

Our method is also related to work that adopts a Lie group formalism in representation learning. Since the seminal work of (Rao & Ruderman, 1998) that proposed learning Lie group generators from dynamic signals, the polar parametrization was explored in (Cohen & Welling, 2014) to identify irreducible representations in a synthetic dataset. The continuous group formalism has also been used in complement to to sparse coding (Chen et al., 2018; Chau et al., 2020) to model natural images as a manifold in the space of pixels. In contrast, our formulation relies on a predictive formulation rather than coding. Most importantly, our work focuses on local symmetries rather than global signal transformations: a polar predictor is able to adaptively capture and exploit different transformations in different regions of complex high-dimensional signals.

Finally, in the fluid mechanics literature, the Koopman operator approach (Mezić, 2005) has been used to lift a system from its original state-space to a higher dimensional representation space where its dynamics can be linearized - a dynamical analog of the well known kernel trick. This formalism has spurred a line of work in machine learning that relies on autoencoders to learn coordinate systems that approximately linearize dynamics (Lusch et al., 2018). In this perspective, our work can also be interpreted as learning the spectral properties of an abstract Koopman operator operating on video data, specifically estimating its complex eigenvectors.

## 5 DISCUSSION

We have presented a self-supervised representation-learning framework based on temporal prediction of image sequences via polar straightening. The optimized mappings validate the idea that predictable features can be extracted from natural video using an angular extrapolation in representation space. Our results also demonstrate that building structure into the architecture can be advantageous, provided that the said structure is well matched to the task. Specifically, we rely on the polar coordinate transformation as a bivariate non-linear activation function to pair together coefficients of the representation and show that such an activation function, all else being equal, induces performance benefits. This exemplifies a fundamental theme in computational vision: when possible, let the representation do the analysis. More concretely, this effect was enabled by bundling angular computations, such that the phase manipulation remains implicit (angular variables are not directly represented in our implementation). This choice of prediction mechanism, which was motivated by principles of signal processing and harmonic analysis, acts as a structural prior and embodies the general hypothesis that evolving content will continue to evolve in the same way. Although the conventional deep convolutional network (CNN) considered here could in principle have discovered this solution, it failed to do so (within the constraints of our setup), which emphasizes the importance of architectural design choices that act as inductive biases. Using terminology from group theory, our polar models factorize signals into an invariant part, which is stable in time, and an equivariant part, which evolves linearly.

This approach to prediction has the advantage of being motion-informed while not relying on explicit motion estimation. Because it is not constrained to assigning a single motion vector at every location and instead relies on a distribution of phases, this method outperforms the motion compensated algorithm (it also runs several order of magnitude faster). The polar predictor is not only able to bypass known difficulties of motion estimation, but also to generate a representation of each frame. It would be interesting to validate the generality of that representation by evaluating it on other tasks like object categorization or segmentation, as well as estimation of heading direction of a moving observer. Several natural extensions of the work presented here can be further explored: (i) treating the angular extrapolation prediction mechanism as a more general building block that is cascaded in a multi-layer architecture; (ii) predicting at longer timescales from representations deeper in the hierarchy; (iii) measuring prediction error directly in the representation domain, while avoiding representation collapse - this would allow a potential connection with biological neural architecture and to human visual perception.

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

## A  APPENDIX

Table 2: Prediction error computed on the DAVIS dataset. Values indicate average PSNR and SSIM. All methods obtain relatively low structural similarity scores (Wang et al., 2004), a perceptual measure of similarity that equals one when both images are identical. This indicates that prediction on this dataset is quite challenging.

| metric | set | Algorithm | | | | | |
|--------|-----|------|------|-------|------|-------|--------|
|        |     | Copy | cMC  | deepL | CNN  | PP    | deepPP |
| PSNR ↑ | train | 21.32 | 23.82 | 23.18 | 23.84 | 24.49 | **24.52** |
|        | test  | 20.06 | 22.37 | 22.30 | 22.82 | **23.46** | 23.35 |
| SSIM ↑ | train | 0.52 | 0.64 | 0.58 | 0.62 | **0.65** | 0.64 |
|        | test  | 0.50 | 0.62 | 0.55 | 0.59 | **0.63** | 0.61 |

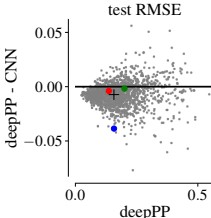

Figure 6: Another comparison - see caption of Fig. 3. The deep polar predictor generally outperforms the CNN over the whole range of difficulties.

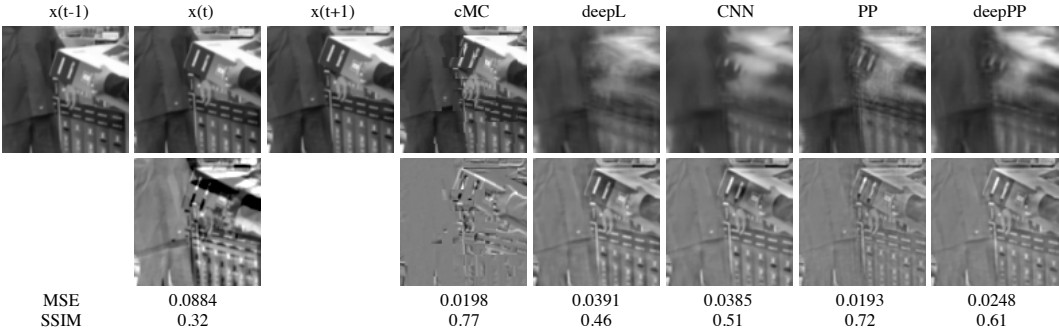

Figure 7: Another example image sequence - see caption of Fig. 4. A video with one portion not moving. In these regions the cMC is most effective, and PP and deepPP outperform CNN and deepL.

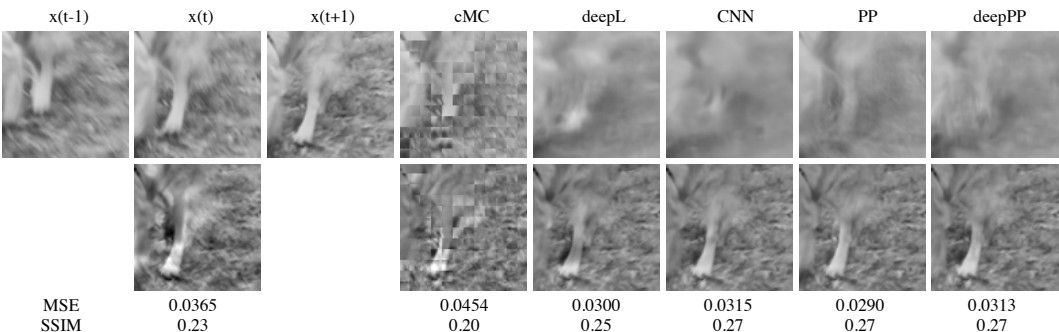

Figure 8: Another image sequence - see caption of Fig. 4. As expected, in presence of large non-rigid motion, with texture and object deformations, prediction is very difficult and none of the tested methods performs well.

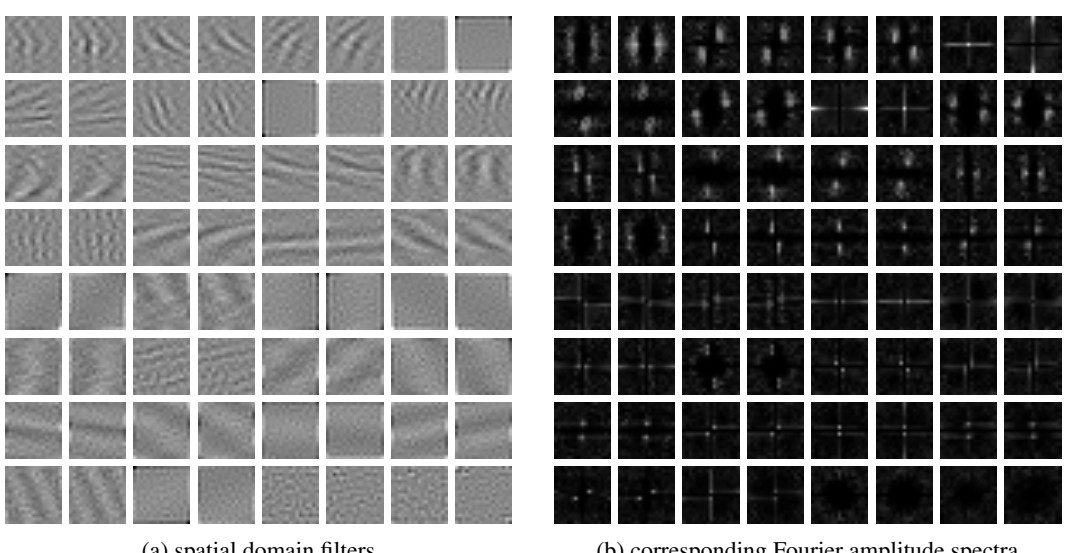

(a) spatial domain filters        (b) corresponding Fourier amplitude spectra

Figure 9: Optimized filters of a polar predictor. The 32 pairs of filters are sorted by their norm and their amplitude spectrum is displayed at corresponding locations on the right panel. Observe that the filters are selective for orientation and spatial frequency, tile the frequency spectrum, and form quadrature pairs. It is interesting that such structured results were obtained from optimizing filters jointly and from a random initial condition, but notice that some of the learned filters do not exactly conform to the idealized description just given.

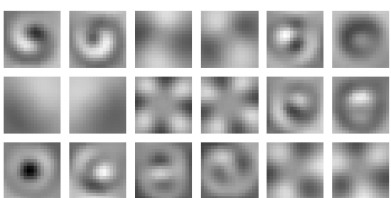

Figure 10: Another set of optimized filters of size $17 \times 17$, these trained in a PP architecture on synthetic sequences generated by rotating image patches.

