# OpenReview forum: "Learning to represent and predict evolving visual signals via polar straightening"
_ICLR.cc/2023/Conference — Submitted to ICLR 2023_

### Official Review · Reviewer_67e4 · 2022-10-23

**Confidence:** 4
**Correctness:** 3
**Technical Novelty And Significance:** 2
**Empirical Novelty And Significance:** 2
**Recommendation:** 5

**Clarity, Quality, Novelty And Reproducibility:**

The paper is generally well written. The idea of the paper is new in the sense that it is trying to develop a new dynamic model that is lightweight and leverages the ideas from signal processing to learn appropriate representation. However, the paper needs some strong improvements in a couple of fronts to be ready for publication. The paper contains some information about reproducibility, although that could also be improved.

**Strength And Weaknesses:**

I appreciate that the paper tries to develop a simple, lightweight and interpretable method and provides insights on the algorithm. Some of the ideas in the paper like learning an appropriate representation is really powerful is an important message.

However, the paper needs strengthening in several fronts. The first and the most important question is a theoretical and a conceptual one. Even though the inspiration from Fourier shift theorem is interesting, it is hard to justify the method based just on that since in real videos, frame to frame transformation is rarely just a translation. Second, even though conceptually the paper discusses Fourier transform, the implementation requires a function (W or g) to be learned which could very well be different from the Fourier transform. However, this limitation can be easily mitigated if the authors choose to develop their work from the perspective of Koopman embedding [1,3,4]. Koopman operator theory is precisely the generalization that authors are looking for. Koopman embedding is the representation space where the complicated nonlinear dynamics are represented as linear dynamics. The dynamics matrix A discussed in the paper could actually be thought as the Koopman operator.

Interestingly, there already exist a couple of papers that try to use Koopman embedding in video prediction. For example, Comas et al [2] is conceptually doing something very similar to this paper, although they are separating objects from the video and applying linear dynamics to the individual objects in some representation space.

Second important front where the paper needs improvement is its experiments or empirical validation. Although some of the insights in the paper are interesting, they need to be validated in a couple of dataset to support the calim that those results and improvements are indeed general. At the moment, the paper only validates results in one dataset.

Third important improvement could come from comparison with stronger deep learning methods in video prediction. It is interesting that the method beats vanilla CNN or deep network without nonlinearity, but there are other specialized works for the purpose of video prediction (e.g. DDPAE [5]). Comparison to those will tell us where the method stands.

References:
1. Bethany Lusch, J. N. Kutz, and S. Brunton. Deep learning for universal linear embeddings of nonlinear dynamics. Nature Communications, 9, 2018.
2. Comas, A., Ghimire, S., Li, H., Sznaier, M. and Camps, O., 2021. Self-Supervised Decomposition, Disentanglement and Prediction of Video Sequences while Interpreting Dynamics: A Koopman Perspective. arXiv preprint arXiv:2110.00547.
3. Omri Azencot, N. Benjamin Erichson, Vanessa Lin, and Michael Mahoney. Forecasting sequential data using consistent Koopman autoencoders. In International Conference on Machine Learning (ICML), 2020
4. Jeremy Morton, F. Witherden, and Mykel J. Kochenderfer. Deep variational koopman models: Inferring koopman observations for uncertainty-aware dynamics modeling and control. In International Joint Conferences on Artificial Intelligence (IJCAI), 2019
5. Hsieh, Jun-Ting, Bingbin Liu, De-An Huang, Li F. Fei-Fei, and Juan Carlos Niebles. "Learning to decompose and disentangle representations for video prediction." Advances in neural information processing systems 31 (2018).

**Summary Of The Paper:**

Inspired by the Fourier Shift Theorem, the paper tries to learn a representation space where the temporal evolution is equivalent to a matrix multiplication by a dynamics matrix A(t). This dynamics matrix is learned by minimizing the one step prediction error in an end to end fashion. More specifically, a frame x(t) is encoded into representation z, where it is advanced through the dynamics matrix and then decoded back. The decoded should be same as the next frame x(t+1). Through experiments on DAVIS dataset, the paper shows that this kind of representation is powerful and outperforms a couple of baselines like convolutional network based prediction and causal motion compensation.

**Summary Of The Review:**

I liked the fresh perspective presented in the paper. However, the paper has some rooms for improvement as discussed earlier. Hence, I am on the side of rejection at the moment.

---

> ### Author Response · Authors · 2022-11-19
> **response to initial review**
>
>
> Thank you for your positive comments on the value of our work as a simple interpretable model, and for your thoughtful questions.
>
> **dynamics matrix is learned by minimizing the one step prediction error**
> The dynamics matrix is not learned, but restricted to angular extrapolation -- only the parameterized mappings are optimized. We assume a constant rate of phase advance, which is motivated by the observation that inertia imbues natural signals with temporal smoothness.
>
> **needs strengthening in theoretical and conceptual front**
> Thank you for flagging this important issue, we had neglected to express the theoretical foundation of our formulation (in terms of compact commutative Lie groups) and have now included this in sub-section 1.3.
>
> **frame to frame transformation is rarely just a translation [...] the implementation requires a function to be learned which could very well be different from the Fourier transform**
> The global Fourier / translation case was meant to provide a motivating base case.  Our models operate on local neighborhoods, and the polar extrapolation of Eq. (2) is performed at each spatial location. But note that in a toy synthetic experiment where basis functions are global and signals are rigidly translating, we have verified that our framework does recover the Discrete Fourier Transform matrix exactly.
>
> **the perspective of Koopman embedding**
> Thank you for bringing this dynamical systems formalism to our attention.
> We did a bit of reading and think that our work can be interpreted as estimating some of the eigenfunctions of a Koopman operator governing evolution of natural videos. We've added a paragraph to the related work section.
>
> **.. needs strengthening in experiments or empirical validation**
> We have bolstered the experimental results by training all models under consideration on a second, much larger, dataset of natural videos (UCF101).  As shown in Table 1, results are consistent with those we originally reported for the DAVIS dataset.
>
> **they are separating objects from the video**
> In natural videos, which are the ultimate test bed, it is difficult to define and separate objects.
>
> **comparison with stronger deep learning methods in video prediction**
> Although we initially intended to make more of these comparisons, it has proven difficult to find usable code online, as authors often do not release the trained weights of their models. If you have a specific suggestion in mind, please do let us know, and we will include comparisons.
>
> **information about reproducibility, although that could also be improved**
> We will release our code (with trained parameters) upon acceptance/publication.

---

> > ### Comment · Reviewer_67e4 · 2022-11-22
> > **Thanks for clarification, but the paper still looks quite confusing**
> >
> > I thank authors for clarification, attempt to improve theoretical justification and addition of related works.
> >
> > Thanks for clarifying that dynamics matrix is just a phase advance following $x(t)-x(t-1)$.
> >
> > I believe that the addition of commutative Lie group is not helping the conceptual framework of the work. It is even more confusing now. The main point of confusion arise from the fact that if you take Fourier transform, then the phase makes sense. If you use an arbitrary neural network, then the transformation is not Fourier but something else. I fail to see how Lie algebra is helping justify the use of nonlinear neural network. Next important question now is, if you are doing phase advance if the operator is Fourier transform, what are you doing now when the operator is convolutional neural network? and why is it a good idea to use temporally local information to do the extrapolation? How does it help in long term prediction?
> >
> > "the polar extrapolation of Eq. (2) is performed at each spatial location"
> > After you have transformed with the CNN operator, there is not spatial location in the transformed space, just like the spatial information is absorbed when you take Fourier transform and what you get out is the Fourier coefficient. They represent frequency components. Now, with CNN operator, it's not clear what it represents. But, it seems like authors use the local temporal smoothness idea for each component separately.
> >
> > Post rebuttal, I do not see paper strengthened. The authors claim that their main contribution is introduction of this new conceptual framework and not to beat state of the art. That is fine, but the conceptual framework is itself quite weak. I will keep my score.

---

> > > ### Author Response · Authors · 2022-11-23
> > > **response and clarifications**
> > >
> > > Thank you for your time and comments.
> > >
> > > **If you use an arbitrary neural network, then the transformation is not Fourier but something else. [...] if you are doing phase advance if the operator is Fourier transform, what are you doing now when the operator is convolutional neural network?**
> > > This the central point of the work: The group representation framework justifies the parametrization of prediction in terms of phase advance in some representation space, and we train a network to find such a space. A shallow network learns localized oriented filters, in conjugate phase-shifted pairs, analogous to the energy/phase operators that have been used to estimate local translations (optic flow) in the computer vision literature (e.g., Fleet \& Jepson, 1990). But the method is capable of learning representations beyond these. For example, on a dataset of rotated image patches, a global (ie. no convolution) polar predictor recovers circular harmonic basis functions (Figure 10 in the appendix). We are working to analyze/interpret what is learned in the deepPP network, which performs better prediction than the shallow PP, but this has so far proven difficult (as with interpretability of Deep Nets in general). For example, prediction near occlusion boundaries is a chronic problem for flow-based algorithms, and although this transformation is not a group action (it is not invertible), we have some evidence that deepPP is succeeding.
> > >
> > > **why is it a good idea to use temporally local information to do the extrapolation? How does it help in long term prediction?**
> > > Long term video prediction is our eventual goal, but the current study is focused/limited on the short-term problem. We do observe that the polar predictor fails gracefully and gradually as we lengthen the prediction intervals.
> > >
> > > **After you have transformed with the CNN operator, there is not spatial location in the transformed space**
> > > There seems to be a misunderstanding here: our architectures are strictly convolutional (localized filters, no downsampling), and there is no fully-connected final stage. Thus, all channels are 2D maps, with their activation values computed from localized receptive fields (for both PP and deepPP, these cover 17x17 pixels).
> > >
> > > **it seems like the authors use temporal smoothness idea for each component separately**
> > > Yes - conceptually, we are attempting to decompose complex transformations into elementary ones (analogous to decomposing global translations using the Fourier decomposition).  Each ``component'' is comprised of a pair of channels whose activations at each location are considered as a complex number, whose phase advances linearly.

---

### Official Review · Reviewer_gEGq · 2022-10-25

**Confidence:** 3
**Correctness:** 3
**Technical Novelty And Significance:** 4
**Empirical Novelty And Significance:** 3
**Recommendation:** 6

**Clarity, Quality, Novelty And Reproducibility:**

Overall I think it is a nice and novel idea that is well motivated and well explained. The paper is well written and easy to follow, but the experiments are somewhat underwhelming due to the points discussed above.

The results can probably be reproduced relatively easily as the experiments are relatively straightforward and well explained. Unfortunately I could not find a statement about availability of code.


**Strength And Weaknesses:**

### Strengths

 + Conceptually well motivated idea
 + Paper is well written and easy to read


### Weaknesses

 1. Only small dataset and shallow architecture
 1. Unclear how robust improvements are


### 1. Small dataset and shallow architecture

It is great to see that the well-motivated inductive bias helps in the relatively data-limited regime the authors study. However, as video data is abundant and easy to obtain in masses, it is not clear to me how relevant such an inductive bias is. Would the advantage also hold on a much larger dataset with a much deeper network architecture?

The fact that the linear encoder/decoder works so well suggests to me that not much of relevance can be learned from such a small dataset. Presumably a deep architecture trained on a large dataset would learn something about objects and their 3d motion patterns and be able to outperform a linear encoder/decoder by a large margin.


### 2. Robustness of improvements

Related to the previous point that the linear method is so good, I think it would be important to provide error bars on Table 1. How confident are we that the differences between the methods are indeed significant? How much variability is there between multiple differently initialized models? What does it tell us that SSIM is more or less the same across cMC, PP and deepPP?


**Summary Of The Paper:**

The authors propose a self-supervised learning objective based on next-frame prediction that uses angular extrapolation in polar coordinates while keeping amplitudes constant. They show that this inductive bias of using angular instead of linear extrapolation improves next-frame prediction performance in terms of MSE and SSIM on video snippets taken from the DAVIS dataset.


**Summary Of The Review:**

Nice and simple paper, but somewhat weak and shallow on the experimental side.

---

> ### Author Response · Authors · 2022-11-19
> **response to initial review**
>
> Thank you for your review and comments.
>
> **as video data is abundant and easy to obtain in masses, it is not clear to me how relevant such an inductive bias is.**
> We agree that, in principle (according to the universal approximation theorem) a large enough neural network trained on a sufficiently big dataset should be able to approximate any function.  This is a sort of upper bound for the entire field of machine learning. But we'd argue that achieving similar performance with a much shallower or architecture is of great importance.  Specifically, smaller architectures
> * are generally more interpretable - we can learn something from simple models that work;
> * are more readily related to or configured to match biology, providing insights about visual neurobiology or perception;
> * are essential for engineered systems, which face severe limitations and constraints in terms of power consumption, size, and robustness (eg. an algorithm that must run in a phone).
>
> **The fact that the linear encoder/decoder works so well...**
> The PP model in Table 1 is not a linear encoder/decoder: the prediction involves angular extrapolation, which is significantly more nonlinear than the rectifiers used in most CNNs (see Equation (2), or (3)).  The high level of performance arises because the polar prediction scheme is well-matched to the spatio-temporal content of the videos.  Note that the DAVIS dataset contains complex natural sequences, unlike moving MNIST (which is synthesized from static digit images), and KTH (which is composed of simple clips captured from a stationary camera, with a blank background).
>
> **It is great to see that the well-motivated inductive bias helps in the relatively data-limited regime the authors study. However, as video data is abundant and easy to obtain in masses, it is not clear to me how relevant such an inductive bias is**
> See Table 1: Our newly added results on the much larger UCF101 dataset (also natural videos) demonstrate that the inductive bias continues to help even when data is abundant.
>
> **How confident are we that the differences between the methods are indeed significant?**
> These large experiments are slow to run, and we do not yet have enough data to provide error bars on the performance estimates. But we feel the replication of the original results on the much larger  UCF101 dataset help build confidence in the original claims of the paper.
>
> **What does it tell us that SSIM is more or less the same across cMC, PP and deepPP?**
> We decided to move the table containing SSIM scores to the appendix because the networks were not trained for this objective and the results do not offer much beyond what is expressed in the MSE values.
>
> **could not find a statement about availability of code.**
> We will release code when the manuscript is accepted/published.

---

### Official Review · Reviewer_zsxR · 2022-10-28

**Confidence:** 2
**Correctness:** 4
**Technical Novelty And Significance:** 3
**Empirical Novelty And Significance:** 3
**Recommendation:** 5

**Clarity, Quality, Novelty And Reproducibility:**

Clarity: The writing is clear, but not being an expert in this area (and not being a EE guy, Fourier transforms are still somewhat mysterious to me!), I'm sure I didn't understand parts of the paper.

Quality: The quality of the results appear to be good. However, not being familiar with the area, I don't know if the systems used as baselines are state of the art. Hopefully other reviewers will know.

Novelty: As far as I know, this is completely novel.

Reproducibility: I assume they will share their code if the paper is accepted.

Typos, wording suggestions:

p1: "For example, in cases of expanding or rotating motion, discontinuous motion at occlusion boundaries, or mixtures of motion arising from semi-transparent surfaces (e.g., viewing the world through a dirty pane of glass)." This sentence no verb.

p4: equation in 4th line from the bottom: In order for the indexes of the y variables to take on values from 1:64, if k is 1:32, then the subscripts for the two instances of y here should be 2k-1 and 2k.

p6, 6 lines up from section 3: You say elsewhere (second-to-last sentence in the bottom full paragraph on page 5 - or is this only for the cMC method?) that you will use frames t-1 and t-2 to predict frame t+1, if I understood this correctly. In that case, with 11 frames, you can only predict 8 frames, not 9. But I guess from elsewhere in the paper that this only applies to cMC, so never mind if so!

Section 3.1: 1st pp, second sentence. I don't understand this sentence - is there a noun missing at the end? half of what?

Same paragraph, 3rd line from the bottom: auto-encoder -> autoencoder

3rd line from the bottom: It would help a bit to note that difficulty increases from left to right in the plots in Figure 3.

Figures 5-8: What is the first image in the second row of these figures? Is that just using x(t) as the prediction? of x(t+1)? Also, in Figure 5, I can't tell what this is a video of. Perhaps a bit of explanation would help? Also, in Figure 6, it looks like you are going backwards in time rather than forwards - the cyclist seems to be moving backwards. The same comment applies to Figure 7 - it looks like the person is moving backwards?

**Strength And Weaknesses:**

Strengths:

+ The inspiration for this method is good.

+ I am not an expert in this area, but it appears to be a completely novel method.

+ The performance is good compared to the baseline methods.

+ The writing is very clear.

Weaknesses, with concrete, actionable feedback

- It is hard to evaluate the qualitative results in Figures 5-8.


**Summary Of The Paper:**

This paper proposes a new way to predict video frames inspired by how rigid objects translate in the Fourier domain, where translation becomes angular progression in a straight line (in polar coordinates). The model simulates (as I understand it) the fourier transform using a particular non-linearity that projects the data into a space similar to the fourier domain, transforms it, and then maps it back. The method is compared to a number of other methods and shows superior performance in terms of MSE and SSIM scores. One version uses orders of magnitude fewer parameters and still outperforms the deep learning baseline methods in terms of MSE. On the other hand, it just barely outperforms a 2 parameter method in terms of SSIM.


**Summary Of The Review:**

To the extent that my knowledge about this area is limited to what is in the paper, this is a novel approach to video prediction, and performs well against the baselines. However, I recall a paper or talk by Yann LeCun and NeurIPS a few years ago that gave very good predictions of video frames by assuming a latent variable that somehow influenced the prediction and "chose" one future out of the possible ones. The predictions he showed were very clear, but I'm not sure how that relates to this work. Also, the straight convolutional network predictor approach is from 2016; surely more has happened in this area in the last 6 years?

---

> ### Author Response · Authors · 2022-11-19
> **response to initial review**
>
> Thank you for your review and comments.
>
> **The model simulates [...] the fourier transform ...**
> We'd like to emphasize that our method does not aim to simulation the Fourier Transform, but rather to generalize it. The Fourier case is provided as motivation for the scheme of ``polar prediction" in the  case of translational motion, but in fact there is a large family of transformations that can be captured by polar prediction in other domains (we now describe this in a newly added subsection - 1.3). Also, the Fourier transform is global, but our model learns a representation on local regions.  In the updated manuscript, we added a sub-section at the end of the introduction to communicate this point more directly.
>
> **On the other hand, it just barely outperforms a 2 parameter method in terms of SSIM.**
> Although the causal motion compensation performs well on the DAVIS dataset, its worth noting that it has a much slower run-time (at least two orders of magnitude) than any other model considered in this study.
> Moreover, this approach performs poorly on UCF101, a new larger dataset of natural videos that we've included in the updated manuscript - see Table 1.
>
> **Also, the straight convolutional network predictor approach is from 2016; surely more has happened in this area in the last 6 years?**
> To our knowledge, most of the solutions in the literature follow the same basic formulation, of predicting a frame from previous frames using a network trained with supervision. Of course, these do vary in architecture, regularization, optimization procedure, dataset, etc, and we had hoped to compare with a more recent method.  But it has proven challenging to find usable code online, as authors often do not release the trained weights of their models. If you have a specific suggestion in mind, please do let us know, and we will include comparisons.
> On the other side of this coin, we will be releasing our code (with trained parameters) upon acceptance/publication.
>
> **Typos**
> Thank you for pointing out typos and unclear wording, these have been fixed in the updated manuscript.
>
> **... use frames t-1 and t-2 to predict frame t+1**
> Please note that all predictions are of frame t+1, given frame t and t-1.
>
> **It is hard to evaluate the qualitative results**
> We've augmented the text to help readers interpret the qualitative results in Figs 5 and 6.
>
> **I recall a paper or talk by Yann LeCun and NeurIPS a few years ago**
> Thank you for mentioning the probabilistic approach to prediction championed by Yann Lecun. Our current framework is purely deterministic, although we have been discussing extensions to incorporate these kind of stochastic or multi-modal behaviors.

---

### Author Response · Authors · 2022-11-19
**General comments**

We have bolstered the paper by adding experiments on a second, commonly used and much larger natural video dataset (UCF101). The new results are consistent with our original claims.

Although we did train and compare our approach to a large deep network, we do not aim to compete with the current state of the art. Indeed, unavailability of trained weights and differences in data pre-processing make these comparisons difficult.

Instead we focus on a conceptual message: describing and demonstrating the benefits of a theoretically grounded non-linearity for temporal prediction.
We designed experiments accordingly and changed one aspect of the architecture at a time so that the reported comparisons are fair and informative.

---

### Decision · Program_Chairs · 2023-01-20

**Decision:**

Reject

**Justification For Why Not Higher Score:**

No real support for acceptance by any of the reviewers.

**Justification For Why Not Lower Score:**

NA

**Metareview: Summary, Strengths And Weaknesses:**

The authors propose a novel approach to next-frame prediction  in video sequences using natural polar coordinates. Compared to a linear approach, angular extrapolation in polar coordinates achieves higher accuracy on video snippets taken from the DAVIS dataset. There was consensus that the proposed framework was interesting but one of the reviewers mentioned a possible connection to Koopman operators which the authors acknowledged in the rebuttal but failed to adequately address. Additional concerns included the fact that the linear baseline performed so well and this point was not adequately addressed in the the rebuttal.


**Summary Of Ac-Reviewer Meeting:**

There was general consensus that the proposed framework makes sense intuitively and the problem formulation in interesting. However, the reviewers were unclear about a possible connection with Koopman operators (which was  brought up by one of the reviewers during the first round of reviews) and there was general agreement that the authors should have addressed this concern better in their rebuttal. Furthermore, the reviewers were not clear about the actual application of the work (and they were not sure the authors knew either). Possible applications include self-supervision and compression but the authors have to choose evaluate on the corresponding benchmarks and compare to SOTA approaches. As it stands the experimental evaluation appears somewhat unconvincing.